# Acid-catalyzed hydrolysis kinetics of organic hydroperoxides: Computational strategy and structure-activity relationship

Qiaojing Zhao[1], Fangfang Ma[1,2], Hui Zhao[1], Qian Xu[1], Rujing Yin[1*], Hong-Bin Xie[1*], Xin Wang[1,3], Jingwen Chen[1]

[1] Key Laboratory of Industrial Ecology and Environmental Engineering (Ministry of Education), School of Environmental Science and Technology, Dalian University of Technology, Dalian 116024, China
[2] College of Resources and Environmental Engineering, Guizhou University, Guiyang 550025, China
[3] Key Laboratory for Semi-Arid Climate Change of the Ministry of Education, College of Atmospheric Sciences, Lanzhou University, Lanzhou 730000, China

*Correspondence to*: Rujing Yin (yinrj@dlut.edu.cn) and Hong-Bin Xie (hbxie@dlut.edu.cn)

**Abstract.** Organic hydroperoxides (ROOHs) are key components of atmospheric aerosols. Determining the acid-catalyzed hydrolysis rate constants ($k_A$) of ROOHs is crucial for assessing their atmospheric fate and environmental impacts. However, available $k_A$ values are limited due to the difficulty in obtaining authentic ROOH standards. Herein, we addressed this limitation by developing a computational strategy and probing the structure-activity relationship of $k_A$ values. We screened the protonated water cluster ($H^+(H_2O)_n$) model, a critical prerequisite for density functional theory (DFT) calculations of $k_A$, by comparing experimental $k_A$ values of four ROOHs with DFT-calculated values using $H^+(H_2O)_n$ ($n = 1, 2, 3, 4$) models. Results show that the $H^+(H_2O)_2$ model reliably predicts $k_A$ values with DFT method. Further investigation of 53 additional ROOHs including 45 model compounds and 8 atmospherically relevant species reveals that substituents at the $C_\alpha$ (the carbon atom directly bonded to the -OOH group) site, including -NH$_2$, -N(CH$_3$)$_2$, -OH, -OCH$_3$, -CH=CH$_2$, -SH, and -PH$_2$, can facilitate acid-catalyzed hydrolysis. Notably, the -NH$_2$ and -N(CH$_3$)$_2$ substituents exhibit stronger facilitating effect than the well-documented -OH and -OCH$_3$ substituents. Additionally, we clarified that not all nitrogen- or oxygen-containing substituents equally enhance $k_A$, as their efficacy depends on the substituents attached to the O or N atoms. This study provides a reliable computational strategy and essential guidelines for predicting $k_A$ values of ROOHs, enabling more accurate simulations in atmospheric chemistry models.

## 1 Introduction

Aerosol liquid water, a crucial constituent of atmospheric aerosols, acts as a reactive medium that enables aqueous-phase chemical transformations (Jin et al., 2020; Su et al., 2022; Shi et al., 2024; Wu et al., 2018). The aqueous-phase transformations of organic compounds substantially modify the physicochemical properties of aerosols, such as chemical composition, optical characteristics, and hygroscopicity, ultimately altering the health and climate effects of aerosols (McNeill, 2015; Herrmann et al., 2015; Zheng et al., 2021; Liu et al., 2023; Lei et al., 2022; Zhang et al., 2024). Recently, aqueous-phase chemistry has gained increasing research interest due to its unique reaction mechanism compared to gas-phase chemistry. However, the

complexity and diversity of these reaction pathways have limited our understanding of aqueous-phase chemistry. The lack of kinetic data for the aqueous-phase chemistry of organic compounds, such as acid-catalyzed hydrolysis or esterification, oxidation, photosensitization, and oligomer processes, further hinders its application in precise simulations of three-dimensional (3-D) atmospheric chemistry models (Ervens et al., 2024; Wieser et al., 2024; Abbatt and Ravishankara, 2023).

Organic hydroperoxides (ROOHs) are ubiquitous in aerosols and predominantly derived from atmospheric oxidation processes involving organic peroxyl radicals (RO$_2$·) or Criegee intermediates (CIs) (Wang et al., 2023). Characterized by one or more hydrophilic hydroperoxide groups (-OOH), ROOHs are highly reactive and usually experience rapid aqueous-phase transformations, forming low-volatile multi-functional species or reactive oxygen species (Enami, 2021; Krapf et al., 2016; Dovrou et al., 2019; Wei et al., 2021; Wei et al., 2022a; Wei et al., 2022b; Wang et al., 2019). Laboratory evidence demonstrates that ROOHs can account for 20%-60% isoprene- or monoterpene-derived secondary organic aerosol (SOA) (Wang et al., 2023; Enami, 2021; Epstein et al., 2014). Therefore, investigating the transformation kinetics of ROOHs is essential for improving our understanding of aqueous-phase chemistry.

Hydrolysis has been identified as an important transformation pathway for ROOHs, yielding hydrogen peroxide (H$_2$O$_2$) and thereby affect atmospheric oxidation capacity (Enami, 2021; Qiu et al., 2019; Qiu et al., 2020a; Hu et al., 2021a; Dai et al., 2024). Recent studies have found that two types of α-substituted ROOHs, α-hydroxyalkyl-hydroperoxides (α-HHs) and α-alkoxyalkyl-hydroperoxides (α-AHs), exhibit rapid hydrolysis under acid-catalyzed conditions. Their lifetimes are short from seconds to minutes at pH<4, with first-order rate coefficients increasing significantly as pH decreases (Qiu et al., 2020b; Hu et al., 2021b; Enami, 2022; Hu et al., 2022; Endo et al., 2022). Two distinct mechanisms have been proposed for the acid-catalyzed hydrolysis of α-HHs and α-AHs. For α-HHs, both the -OOH and -OH groups attached to the same carbon atom participate in the reaction simultaneously, yielding the aldehyde and H$_2$O$_2$ (Qiu et al., 2020b); while α-AHs decompose directly to release H$_2$O$_2$ and generate the carbocation intermediates, which are rapidly hydrated to form the corresponding alcohols (Hu et al., 2021a). Moreover, a latest study further demonstrated that some monoterpene-derived α-acyloxyalkyl-hydroperoxides also decay significantly at low pH (Chang et al., 2025). This highlights the significance of acid-catalyzed hydrolysis in transforming ROOHs and modifying the oxidation capacity of the surrounding atmosphere. However, the currently available data on acid-catalyzed hydrolysis rate constants ($k_A$) are still very limited, impeding a comprehensive understanding of this aqueous-phase transformation process. The $k_A$ data lack manifests in two aspects: i) current research on α-substituted ROOHs is confined to those with hydroxyl, alkoxy, and acyloxy substituents, while the $k_A$ values of α-substituted ROOHs with other substituents remains unclear; ii) despite recent studies have indicated that non-α-substituted ROOHs, which are more abundant in the atmosphere, have longer lifetimes up to days in aqueous phase, detailed investigations for their $k_A$ values are lacking (Dai et al., 2024; Zhao et al., 2022). Therefore, a comprehensive investigation of $k_A$ values for structurally diverse ROOHs in atmosphere is urgently needed to advance our understanding of this aqueous-phase transformation process.

Acquiring $k_A$ values for diverse ROOHs through laboratory experiments is significantly challenging due to the lack of commercial standards and the prerequisite synthesis of target compounds. Given these experimental limitations and the structural diversity of ROOHs, a viable approach is to use the quantum chemical method to conduct structure-activity

relationship investigation. However, previous DFT calculation studies using the simplest protonated water cluster $H_3O^+$ (i.e., $H^+(H_2O)_1$) model have overestimated hydrolysis rates (Hu et al., 2022), since $H_3O^+$ would cluster with water molecules in solution and reduce the proton activity (Agmon et al., 2016). Hence, it is urgent to screen a more suitable protonated water cluster model within the DFT calculation strategy to accurately and efficiently predict the $k_A$ values of ROOHs.

In this study, we initially selected four types of ROOHs with experimental $k_A$ values as tested compounds to search for a suitable protonated water cluster model. Utilizing the screened protonated water cluster model and DFT calculations, the acid-catalyzed hydrolysis of additional 45 ROOH model compounds was systematically investigated to elucidate the structure-activity relationship of $k_A$ values and the corresponding quantitative structure-activity relationship (QSAR) models were developed. These model compounds were selected mainly considering the diverse structures at the $C_\alpha$ (the carbon atom directly bonded to the -OOH group) and $C_\beta$ (the carbon atom attached to $C_\alpha$) sites. The investigation was further extended to other eight atmospheric ROOHs to validate the structure-activity relationship revealed by the model compounds. The screened protonated water cluster model and the revealed structure-activity relationship can be instrumental in predicting the $k_A$ values of atmospheric ROOHs. The established kinetic database for ROOHs would further improve the predictions of 3-D chemical models and deepen our understanding of atmospheric aqueous chemistry.

## 2 Computational details

### 2.1 Electronic structure calculations

All structure optimizations and energy calculations were performed using the Gaussian 09 program (Frisch et al., 2009). Geometry optimization and frequency calculation for the reactants (R), products (P), transition states (TS), pre-reactive complexes (RC), intermediates (IM), and post-reactive complexes (PC) were conducted at the M06-2X/6-31+G(d,p) level of theory (Zhao and Truhlar, 2008). Single-point energy (SPE) calculations were performed at the M06-2X/6-311++G(3df,2pd) level. The employed quantum chemistry calculation method can be succinctly represented as M06-2X/6-311++G(3df,2pd)//M06-2X/6-31+G(d,p). Previous studies have demonstrated that the M06-2X method is suitable for predicting energies and kinetics for the aqueous-phase reactions of organic compounds with a good balance between accuracy and computational efficiency (Ji et al., 2020; Shi et al., 2024; Zhang et al., 2022; Piletic et al., 2013). The solvation model based on density (SMD) was applied to account for the effects of solvent water molecules in the aqueous phase (Marenich et al., 2009). Intrinsic reaction coordinate calculations were used to confirm that the well-defined TSs connect with the corresponding reactants and products (Fukui, 1981). For reaction pathways where TSs could not be successfully located, relaxed scan methods were employed to obtain free-energy surfaces (Zhao et al., 2023; Ryu et al., 2018). Gibbs free energy ($G$) values for each structure at 298.15 K were calculated by combining the SPE with the Gibbs correction energy calculated at the theoretical level of geometry optimization. A correction factor of 1.89 kcal mol$^{-1}$ was applied to the activation free energy ($\Delta G^{\ddagger}$) and reaction free energy ($\Delta_r G$) calculations for reactions where the number of molecules decreases or increases by one from R to TS or from R to P to explain the free energy change from the gas phase standard state of 1 atm to the aqueous phase

standard state of 1 mol $L^{-1}$ (Sadlej-Sosnowska, 2007; Zhang et al., 2015). The combined use of Molclus 1.9.9.9 and Gaussian 09 programs was applied to search for the global minimum of the reactants (Lu., 2022).

## 2.2 $H^+(H_2O)_n$ model selection

A previous study found that the simplest protonated water cluster $H^+(H_2O)_1$ model overestimated acid-catalyzed hydrolysis rates of ROOHs by combining with feasible DFT method (Hu et al., 2022). This is probably because $H^+(H_2O)_1$ clusters with water molecules in the aqueous solution and the proton activity decreases (Agmon et al., 2016). The protonated water cluster model has been found to predict the acid-catalyzed hydrolysis kinetics of other organic compounds such as epoxydiols with high accuracy (Piletic et al., 2013). Here, we screened an appropriate protonated water cluster model by comparing calculated $k_A$ values for reactions between $H^+(H_2O)_n$ ($n$ = 1, 2, 3, 4) and the selected ROOHs with experimental values. Specially, $C_{13}$ α-AH, $C_{12}$ α-AH$_{(1)}$, $C_{12}$ α-AH$_{(2)}$, and $C_{10}$ α-HH were chosen due to the available experimental kinetic data under different pH values (Qiu et al., 2020b; Enami, 2022; Hu et al., 2022). The structures of these four ROOHs and $H^+(H_2O)_n$ ($n$ = 1, 2, 3, 4) are presented in Figs. S1 and S2.

## 2.3 Reaction rate constants calculation

The second-order reaction rate constants $k_A$ of the elementary reactions in acid-catalyzed hydrolysis pathways were calculated using transition state theory (Eq. (1)) (Zhao et al., 2023; Xu et al., 2019). The pseudo-first-order rate constants ($k'_A$) were calculated by combining $k_A$ with the concentration of protonated water clusters, which are determined by pH. Corresponding acid-catalyzed hydrolysis lifetimes ($\tau_{1/e}$) can be subsequently derived by $\tau_{1/e} = 1/k'_A$.

$$k_A = \sigma \frac{k_B T}{h} \exp^{(-\frac{\Delta G^\ddagger}{RT})} , \tag{1}$$

where $\sigma$ is the reaction path degeneracy, T is the temperature (298.15 K), $k_B$ is the Boltzmann constant (J $K^{-1}$), h is the Planck constant (J s), R is the gas constant (8.314 J $mol^{-1}$ $K^{-1}$), $\Delta G^\ddagger$ is the activation free energy.

### 2.4 QSAR modeling

The QSAR models were developed for $k_A$ of α- and β-substituted compounds, respectively, among the selected 45 ROOH model compounds. The Taft ($\sigma^*$) constants, which are widely used to quantify the inductive effects of substituents in aliphatic systems (Lee and von Gunten, 2012; Kim and Huang, 2021; Ra et al., 2025), were employed to correlate with the logarithms of $k_A$ (log $k_A$) values in this study. The $\sigma^*$ values for different substituents, as shown in Table S1, were obtained from the literature (Perrin et al., 1981). Herein, the total effects were evaluated by summing the $\sigma^*$ values ($\sum\sigma^*$) of each substituent (except the -OOH group) at the reaction center, i.e., the $C_\alpha$ site (Table S2).

 **3 Results and discussion**

**3.1 Selection of suitable $H^+(H_2O)_n$ model**

We investigated the acid-catalyzed hydrolysis of four ROOHs, i.e., $C_{13}$ α-AH, $C_{12}$ α-AH$_{(1)}$, $C_{12}$ α-AH$_{(2)}$, and $C_{10}$ α-HH, using $H^+(H_2O)_n$ ($n$ = 1, 2, 3, 4) models with the DFT method. The calculated free-energy profiles for these reactions are presented in Figs. S3-S6. Concluded from the reaction profiles, a two-step acid-catalyzed hydrolysis pathway is revealed for these four ROOHs, with the first step being the rate-limiting step. As shown in Fig. S7, the $H^+$ of $H^+(H_2O)_n$ attacks the -OOH group of the selected four compounds, leading to $C_α$-O bond rupture and simultaneous formation of a carbocation intermediate and $H_2O_2$ in the first step. Subsequently, the carbocation reacts with water molecules and yields the corresponding protonated alcohols.

Using the activation free energies of the rate-limiting step, the pseudo-first-order rate constants $k'_A$ were calculated over a pH range of 0-14 using the M06-2X/6-311++G(3df,2pd)//M06-2X/6-31+G(d,p) method. The calculated $k'_A$ values for the four selected ROOHs with $H^+(H_2O)_n$ models are presented in Fig. 1, along with the experimentally determined hydrolysis rates at different pH values (Qiu et al., 2020b; Enami, 2022; Hu et al., 2022; Endo et al., 2022). The $H^+(H_2O)_2$ model was found to best reproduce the experimental data among $H^+(H_2O)_n$ ($n$ = 1, 2, 3, 4) models. In most cases, the discrepancies between the calculated and the experimental $k'_A$ values are within a factor of 0.5 to 2.2. Two exceptions are $C_{10}$ α-HH at pH 3.3 and $C_{12}$ α-AH$_{(2)}$ at pH 4.5, but their uncertainties remain within one order of magnitude, an acceptable error range. In contrast, the $H^+(H_2O)_4$, $H^+(H_2O)_3$, and $H^+(H_2O)_1$ models result in much larger discrepancies, with errors reaching $1.0 \times 10^{-5}$, $1.1 \times 10^{-3}$, and $2.1 \times 10^8$ times, respectively. Therefore, the computational strategy combining the $H^+(H_2O)_2$ model with the M06-2X/6-311++G(3df,2pd)//M06-2X/6-31+G(d,p) method is well-suited for predicting $k_A$ values of ROOHs.

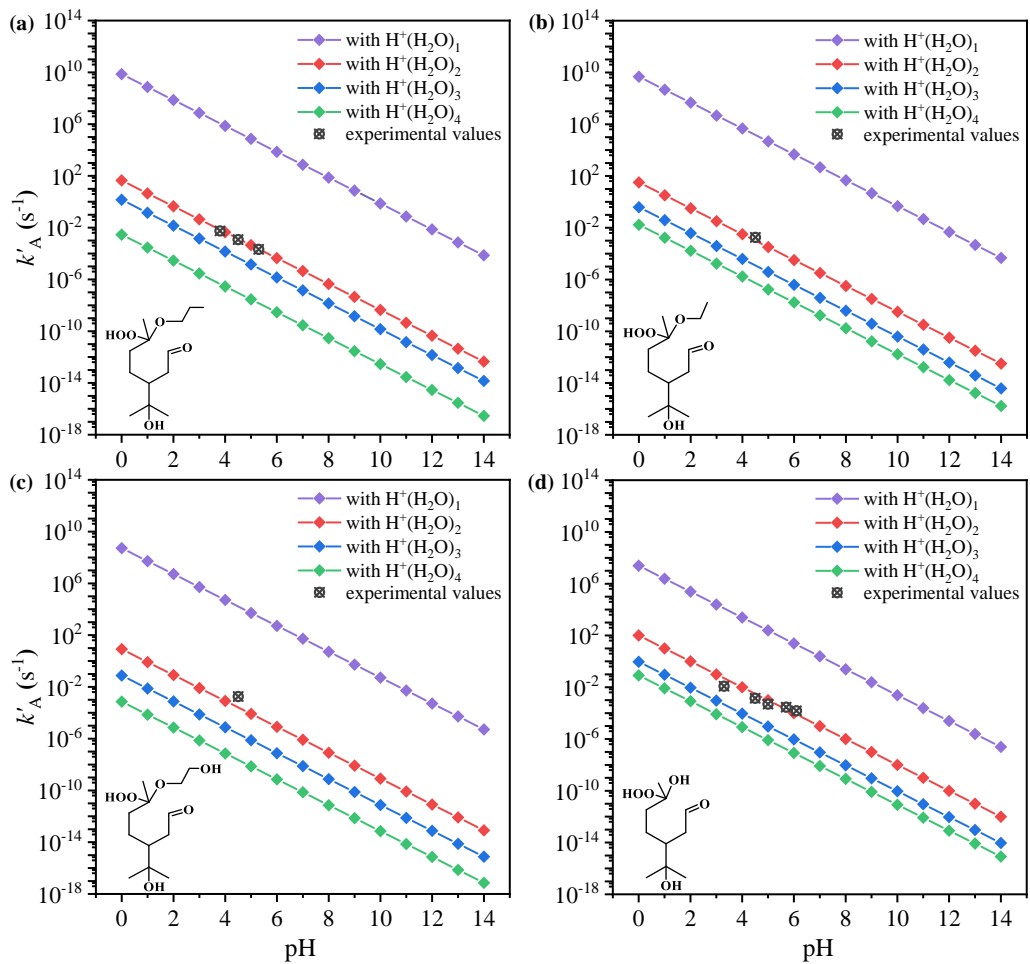

**Figure 1.** Variation of calculated pseudo-first-order acid-catalyzed hydrolysis rate constants ($k'_A$) for four ROOHs along with the available experimental values (Qiu et al., 2020b; Enami, 2022; Hu et al., 2022; Endo et al., 2022). (a) $C_{13}$ α-AH, (b) $C_{12}$ α-AH$_{(1)}$, (c) $C_{12}$ α-AH$_{(2)}$, and (d) $C_{10}$ α-HH.

### 3.2 Structure-activity relationship of acid-catalyzed hydrolysis rate constants

Employing the $H^+(H_2O)_2$ model and the M06-2X/6-311++G(3df,2pd)//M06-2X/6-31+G(d,p) method, we conducted a systematic investigation for structure-activity relationships of $k_A$ values across 45 ROOH model compounds (Fig. 2a). We selected the simplest unsubstituted primary, secondary, and tertiary ROOHs, i.e., $CH_3CH_2OOH$, $CH(CH_3)_2OOH$, and $C(CH_3)_3OOH$, as the reference compounds to assess the influence of substituents. Specifically, the influence of substituents was examined by comparing the $k_A$ values of α-substituted primary, secondary, and tertiary ROOHs, as well as β-substituted tertiary ROOHs, represented as $CH_2(X)OOH$, $CH(CH_3)(X)OOH$, $C(CH_3)_2(X)OOH$, and $C(CH_3)_2(CH_2(X))OOH$, with the corresponding unsubstituted ones, respectively. Here, $X$ denotes substituents including $-NH_2$, $-OH$, $-OCH_3$, $-CH=CH_2$, $-SH$, $-PH_2$, $-F$, $-Cl$, and $-CHO$, covering the major functional group types found in atmospheric ROOHs. Therefore, the effects of

different carbon skeletons at the $C_\alpha$ site (primary, secondary, and tertiary) and different substituents at the $C_\alpha$ and $C_\beta$ sites of the -OOH group on the $k_A$ values of ROOHs were revealed.

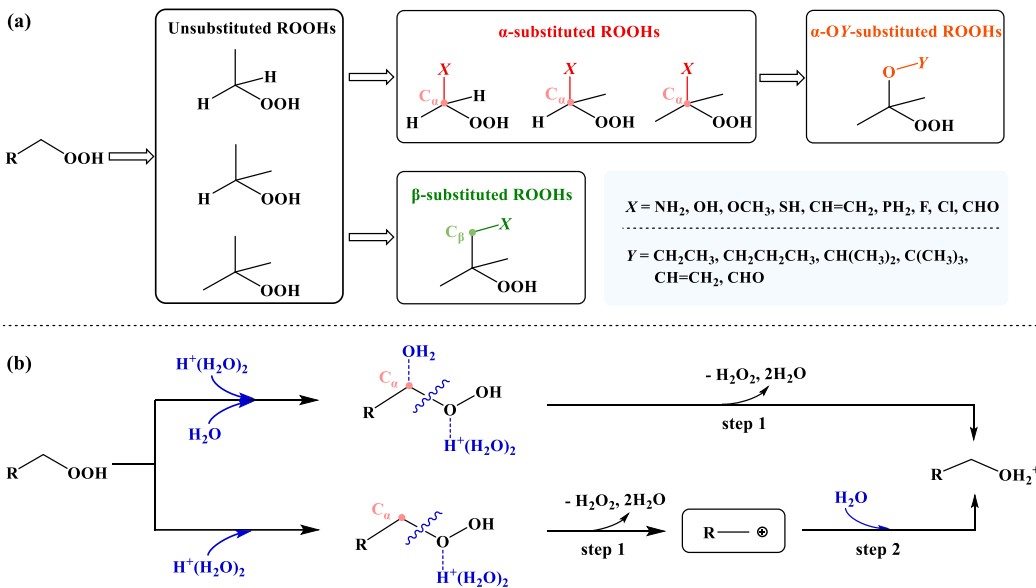

Figure 2. Molecular structure and acid-catalyzed hydrolysis pathways of ROOH model compounds. (a) Molecular structure of 45 ROOH model compounds; (b) One-step and two-step acid-catalyzed hydrolysis pathways of ROOHs.

For the selected 45 ROOH model compounds, their reaction free-energy profiles were calculated and shown in Figs. S8-S12. They follow two distinct reaction pathways, which are simplified and illustrated in Fig. 2b. $CH_3CH_2OOH$, $CH_2(X)OOH$, and $CH(CH_3)(X)OOH$ ($X$ = F, Cl, CHO) undergo a one-step reaction pathway, where $H^+(H_2O)_2$ and an $H_2O$ molecule simultaneously attack the -OOH group and the $C_\alpha$, respectively, leading to the formation of protonated alcohols and $H_2O_2$. Meanwhile, the other ROOH model compounds follow the two-step reaction pathway via the formation of a carbocation intermediate, similar to that of $C_{13}$ α-AH, $C_{12}$ α-AH$_{(1)}$, $C_{12}$ α-AH$_{(2)}$, and $C_{10}$ α-HH. It is found that these model compounds share nearly identical reaction sites (i.e., the -OOH group and its adjacent $C_\alpha$ atom) and follow analogous reaction pathways to $C_{13}$ α-AH, $C_{12}$ α-AH$_{(1)}$, $C_{12}$ α-AH$_{(2)}$, and $C_{10}$ α-HH. This mechanistic similarity implies that despite variations in their carbon skeletons and substituents, protonation and subsequent reactions are primarily localized at the $C_\alpha$-OOH site, which is a common structural feature among ROOHs. Therefore, even though experimental data for these model compounds are currently lacking to validate the computational results, we believe the screened protonated water cluster model can be reasonably extended to investigate the acid-catalyzed hydrolysis of diverse ROOHs with acceptable uncertainty.

Based on reaction activation free energies, the $k_A$ values for 45 ROOH model compounds were calculated, as well as $k'_A$ values under different pH conditions (Table S3 and Fig. 3). As shown in Fig. 3a, the $k'_A$ of tertiary ROOH is the highest among the three unsubstituted ROOHs at the same pH, followed by secondary and primary ROOHs, aligning with the previous research findings (Hu et al., 2022). Furthermore, to clarify the effects of substituent types, we calculated the enhancement factors (EF) for $k'_A$ of substituted ROOHs relative to their unsubstituted reference compounds. Because the predicted $k'_A$ values

160

for ROOHs exhibit a linear dependence on pH, the resulting EF values remain constant across different pH values. These EF values for α-substituted ROOH model compounds, $CH_2(X)OOH$, $CH(CH_3)(X)OOH$, and $C(CH_3)_2(X)OOH$ are shown in Fig. 3b-d, with $CH_3CH_2OOH$, $CH(CH_3)_2OOH$, and $C(CH_3)_3OOH$ as reference compounds, respectively. Substituents such as $-NH_2$, $-OH$, $-OCH_3$, $-SH$, and $-CH=CH_2$ increase $k'_A$, with EF values ranging from $5.1 \times 10^3$ - $6.4 \times 10^{23}$ for primary, $1.3 \times 10^5$ - $5.0 \times 10^{18}$ for secondary, and $1.4$ - $5.7 \times 10^{13}$ for tertiary ROOHs. The $-PH_2$ substituent shows a divergent effect, increasing $k'_A$ of

primary and secondary ROOHs with the EF values of $1.3 \times 10^5$ and $2.5$, respectively, but decreasing that of tertiary ones. While $-OH$ and $-OCH_3$ are known to facilitate the acid-catalyzed hydrolysis of ROOHs (Hu et al., 2021a; Qiu et al., 2020b; Hu et al., 2021b; Enami, 2022; Hu et al., 2022; Endo et al., 2022), our study is the first to show that $-NH_2$, $-SH$, $-CH=CH_2$, and $-PH_2$ can also enhance it. Among them, $-NH_2$ has the highest EF, followed by $-OH$ and $-OCH_3$, $-SH$, $-CH=CH_2$, and $-PH_2$. In contrast, $-F$, $-Cl$, and $-CHO$ significantly decrease the $k'_A$.

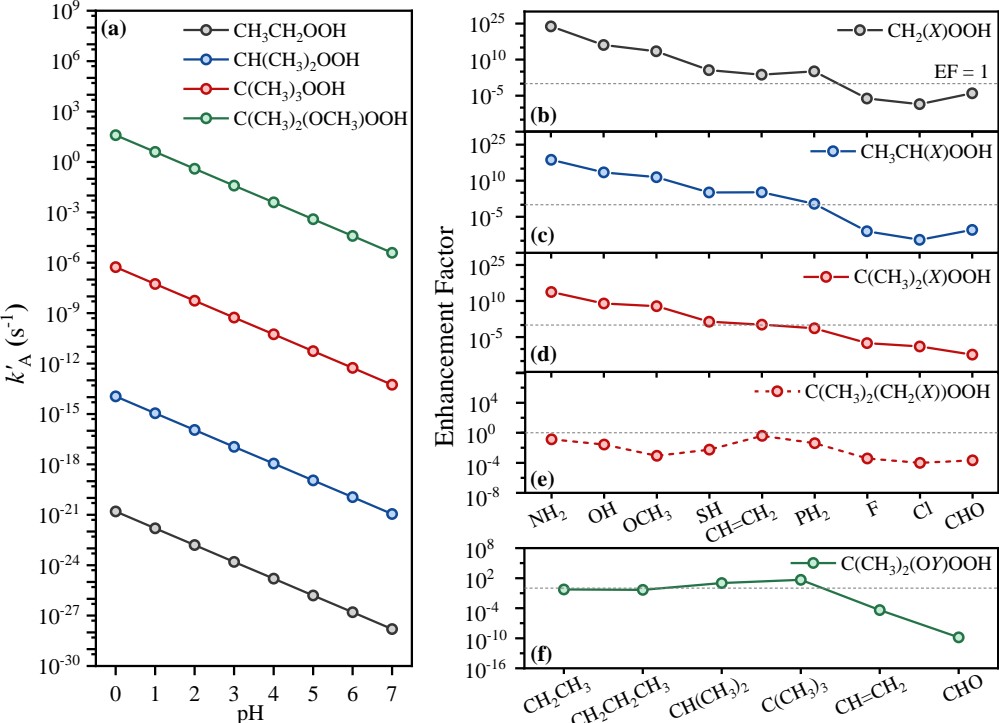

**Figure 3. Calculated pH-dependent pseudo-first-order rate constants ($k'_A$) of ROOH model compounds and substituent enhancement effects on $k'_A$. (a) Calculated $k'_A$-pH profiles for reference ROOHs: $CH_3CH_2OOH$, $CH(CH_3)_2OOH$, $C(CH_3)_3OOH$, and $C(CH_3)_2OCH_3OOH$. (b-f) Enhancement factors (EF = $k'_{A, ROOHs}/k'_{A, reference}$) for (b) $CH_2(X)OOH$ vs. $CH_3CH_2OOH$, (c) $CH(CH_3)(X)OOH$ vs. $CH(CH_3)_2OOH$, (d) $C(CH_3)_2(X)OOH$ vs. $C(CH_3)_3OOH$, (e) $C(CH_3)_2(CH_2(X))OOH$ vs. $C(CH_3)_3OOH$, and (f)**

**$C(CH_3)_2(OY)OOH$ vs. $C(CH_3)_2OCH_3OOH$.**

To further evaluate the effectiveness of acid-catalyzed hydrolysis for these α-substituted model compounds in the atmosphere, we calculated their $\tau_{1/e}$ values at pH 3.8 and 0.9, representing typical aerosol conditions in inland China and the Southeastern United States, respectively (Wang et al., 2022; Jia et al., 2018; Ding et al., 2019; Shi et al., 2019; Xu et al., 2020; Wang et al., 2020; Guo et al., 2015). Compared to the typical atmospheric retention time of ambient aerosols with

approximately 1-2 weeks (Hodzic et al., 2016; Kristiansen et al., 2016), Table 1 shows the calculated $k_A$ and $\tau_{1/e}$ values of the α-substituted ROOH model compounds that undergo effective acid-catalyzed hydrolysis on this time scale at pH 3.8 and 0.9. The $\tau_{1/e}$ values for -NH$_2$-substituted primary ROOH, -NH$_2$- and -OH-substituted secondary ROOHs, as well as -NH$_2$-, -OH-, and -OCH$_3$-substituted tertiary ROOHs range from less than 1 second to 5.5 hours at pH 3.8. As pH decreases to 0.9, effective hydrolysis occurs for -OH-substituted primary ROOH, -OCH$_3$-substituted secondary ROOH, and -SH-substituted tertiary ROOH, with $\tau_{1/e}$ ranging from 40.9 min to 6.8 d. The $\tau_{1/e}$ values for other compounds exceeding 2 weeks are shown in Table S3, indicating the limited effectiveness of acid-catalyzed hydrolysis under typical atmospheric conditions. In summary, substituents such as -NH$_2$, -OH, and -OCH$_3$ can facilitate effective acid-catalyzed hydrolysis of ROOHs under typical ambient conditions, while the -SH substituent is only effective under exceptionally low pH. Although -OH and -OCH$_3$ substituents are well-documented, we show that the -NH$_2$ substituent leads to significantly shorter $\tau_{1/e}$ for the corresponding ROOHs. Despite the -NH$_2$ group prefers to exit in its protonated form (-NH$_3^+$) in the aqueous phase, the rapid reaction of ROOHs with unprotonated -NH$_2$ group may shift the equilibrium -NH$_3^+$ + H$_2$O $\leftrightarrow$ -NH$_2$ + H$_3$O$^+$ forward. Recent studies (Ji et al., 2020; Shi et al., 2024) have reported fast reactions between small α-dicarbonyls and amines or ammonia under acidic aqueous conditions, with corresponding products detected experimentally. This indicates that the ROOHs with unprotonated -NH$_2$ group can still participate in the reaction. In addition, Enami et al. (2010) found that trimethylamine remains largely unprotonated at the air-water interface even at pH 4.0. These findings inspire us that both aqueous-phase and interfacial transformations of α-NH$_2$-substituted ROOHs are plausible. Previous studies demonstrate that α-NH$_2$-substituted ROOHs and their analogues α-N(CH$_3$)$_2$-substituted ones are primarily formed via the reactions of CIs with ammonia or the oxidation of tertiary amines (Li et al., 2024; Kjaergaard et al., 2023; Ma et al., 2021). Therefore, investigating the transformation of these nitrogen-containing ROOHs should be important, particularly in regions with high amine and ammonia concentrations.

Table 1. Calculated acid-catalyzed hydrolysis second-order reaction rate constants ($k_A$) and corresponding lifetimes ($\tau_{1/e}$) of 14 ROOH model compounds under two selected scenarios, Inland China (pH 3.8) and Southeastern United States (pH 0.9).

| Compounds | Formulas | $k_A$ (L mol$^{-1}$ s$^{-1}$) | $\tau_{1/e,\ pH\ 3.8}$ | $\tau_{1/e,\ pH\ 0.9}$ |
|---|---|---|---|---|
| CH$_2$($X$)OOH | CH$_2$(NH$_2$)OOH | $1.01 \times 10^3$ | 6.3 s | < 1 s |
| | CH$_2$(OH)OOH | $1.94 \times 10^{-5}$ | — | 4.7 d |
| CH$_3$CH($X$)OOH | CH$_3$CH(NH$_2$)OOH | $5.60 \times 10^4$ | < 1 s | < 1 s |
| | CH$_3$CH(OH)OOH | $3.20 \times 10^{-1}$ | 5.5 h | 24.8 s |
| | CH$_3$CH(OCH$_3$)OOH | $3.24 \times 10^{-3}$ | — | 40.9 min |
| C(CH$_3$)$_2$($X$)OOH | C(CH$_3$)$_2$(NH$_2$)OOH | $3.10 \times 10^7$ | < 1 s | < 1 s |
| | C(CH$_3$)$_2$(OH)OOH | $4.87 \times 10^2$ | 13.0 s | < 1 s |
| | C(CH$_3$)$_2$(OCH$_3$)OOH | $3.93 \times 10^1$ | 2.7 min | < 1 s |
| | C(CH$_3$)$_2$(SH)OOH | $1.36 \times 10^{-5}$ | — | 6.8 d |
| C(CH$_3$)$_2$(O$Y$)OOH | C(CH$_3$)$_2$(OCH$_2$CH$_3$)OOH | $2.17 \times 10^1$ | 4.9 min | < 1 s |
| | C(CH$_3$)$_2$(OCH$_2$CH$_2$CH$_3$)OOH | $1.76 \times 10^1$ | 6.0 min | < 1 s |
| | C(CH$_3$)$_2$(OCH(CH$_3$)$_2$)OOH | $4.16 \times 10^2$ | 15.2 s | < 1 s |
| | C(CH$_3$)$_2$(OC(CH$_3$)$_3$)OOH | $1.81 \times 10^3$ | 3.5 s | < 1 s |
| | C(CH$_3$)$_2$(OCH=CH$_2$)OOH | $1.62 \times 10^{-3}$ | — | 1.4 h |

It is intriguing to discuss why substituents such as -NH$_2$, -OH, -OCH$_3$, -CH=CH$_2$, -SH, and -PH$_2$ can enhance the acid-catalyzed hydrolysis of ROOHs. As discussed above, a common feature for the acid-catalyzed hydrolysis of these substituted ROOHs is the formation of carbocation intermediates during their two-step reaction pathway. As illustrated in Fig. 4, the greater the enhancing potential of a substituent, the lower the reaction free energy required for carbocation intermediate formation, resulting in higher intermediate stability. This trend remains consistent when using the other computational method, despite differences in the absolute free energies (Fig. S13). Furthermore, carbocation hydration shows minimal impact on reaction energetics, as demonstrated by comparing results from the monohydrated system to those from the implicit solvation model alone, using the reaction of C(CH$_3$)$_2$(OCH$_3$)OOH as a test case (Fig. S14). These findings suggest that the stabilizing effect of these substituents on carbocation intermediates is the driving force behind the enhanced acid-catalyzed hydrolysis. Exceptions for the -OH- and -OCH$_3$-substituted ROOHs should be caused by the slight difference in their two-step reaction pathways (Fig. S7). According to electronic effect theory (Naredla and Klumpp, 2013; Olah, 2001), substituents -NH$_2$, -OH, -OCH$_3$, -SH, -CH=CH$_2$, and -PH$_2$ stabilize carbocation intermediates through conjugated electron donation into the unoccupied orbital of the carbocation. This stabilizing effect follows the order: -NH$_2$ > -OH ≈ -OCH$_3$ > -SH ≈ -CH=CH$_2$ ≈ -PH$_2$. This order aligns with the observed trend in $k_A$ values, where α-NH$_2$-substituted ROOHs exhibit higher $k_A$ values than α-OH- and -OCH$_3$-substituted ones, which in turn are higher than α-SH-, -CH=CH$_2$-, and -PH$_2$-substituted ones (Table S3). Similarly, the stabilizing effect of these substituents on carbocation intermediates can also explain the order of $k_A$ values for tertiary ROOHs > secondary ROOHs > primary ROOHs. Tertiary ROOHs exhibit higher $k_A$ values due to stronger hyperconjugation interactions between the unoccupied p orbital of the carbocation and the additional C-H σ-bonds from the methyl group, which stabilize the carbocation intermediates (Alamiddine and Humbel, 2013).

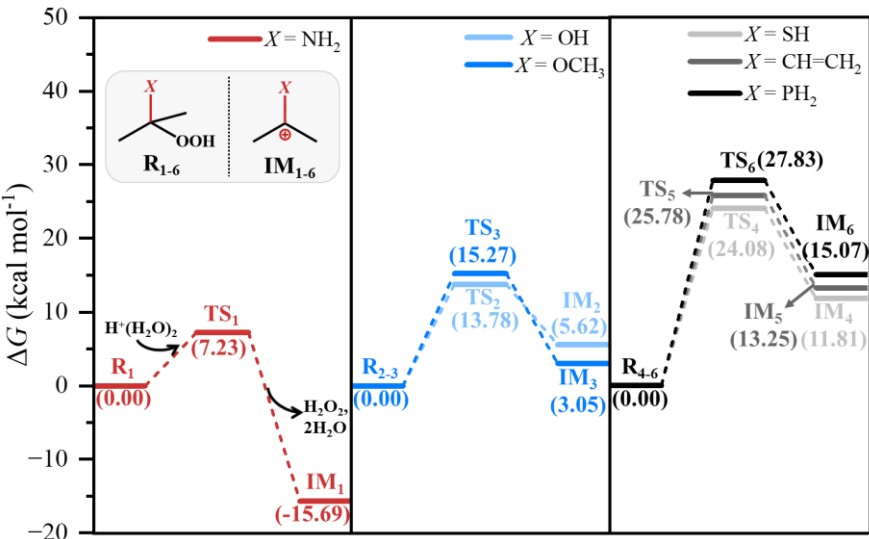

**Figure 4. Calculated schematic free-energy surfaces for carbocation formation during acid-catalyzed hydrolysis of C(CH$_3$)$_2$($X$)OOH. Substituents $X$ denotes NH$_2$, OH, OCH$_3$, CH=CH$_2$, SH, and PH$_2$. Free energies are calculated at the M06-2X/6-311++G(3df,2pd)//M06-2X/6-31+G(d,p) level with the SMD solvation model, with ROOH and H$^+$(H$_2$O)$_2$ set as the reference state (R) at 0 kcal mol$^{-1}$. Transition states (TS) and intermediates (IM) are labeled.**

Based on the above analysis, we deduced that the presence of N and O atoms is the main reason for the strongest facilitating effect of -NH$_2$, -OH, and -OCH$_3$ in the acid-catalyzed hydrolysis of ROOHs. This raises an interesting question of how do substituents attached to N or O atoms affects the $k'_A$ values. To explore this, we examined the effect of substituents attached to O atom as a test case and calculated the EF for $k'_A$ of C(CH$_3$)$_2$(O$Y$)OOH ($Y$ being substituents) compared to the reference compound C(CH$_3$)$_2$OCH$_3$OOH. As shown in Fig. 3f, substituents -CH$_2$CH$_3$ and -CH$_2$CH$_2$CH$_3$ lead to only a minor reduction for $k'_A$. This is validated by the similar experimental $k'_A$ of C$_{12}$ α-AH$_{(1)}$ and C$_{13}$ α-AH, which possess -OCH$_2$CH$_3$ and -OCH$_2$CH$_2$CH$_3$, respectively (Hu et al., 2022). Substituents -CH(CH$_3$)$_2$ and -C(CH$_3$)$_3$ increase the $k'_A$, with EF values of 10.6 and 45.9, respectively. However, -CH=CH$_2$ and -CHO substituents dramatically decrease $k'_A$ by approximately 5 and 10 orders of magnitude, respectively. As shown in Table 1, the C(CH$_3$)$_2$(O$Y$)OOH with $Y$ being four alkyl substituents undergo rapid acid-catalyzed hydrolysis with $\tau_{1/e}$ values ranging from 3.5 s to 6.0 min even at pH 3.8, while C(CH$_3$)$_2$(OCH=CH$_2$)OOH only undergoes effective reaction at pH 0.9 ($\tau_{1/e}$ = 1.4 h). In contrast, the acid-catalyzed hydrolysis of C(CH$_3$)$_2$(OCHO)OOH is difficult to occur. These findings underscore that not all oxygen-containing substituents at the C$_\alpha$ site are equally effective in promoting acid-catalyzed hydrolysis of ROOHs, highlighting the importance of specific structural features. The substituent effects on the O atom could be reasonably extrapolated to the N atom. It should be noted that although the direct acid-catalyzed hydrolysis of C(CH$_3$)$_2$(OCHO)OOH is impossible, the indirect reaction initiated by the hydrolysis of the -OCHO group has been proposed (Chang et al., 2025; Zhao et al., 2018), which is beyond the range of this study.

For the β-substituted ROOHs, tertiary C(CH$_3$)$_2$(CH$_2$($X$))OOH ($X$ = -NH$_2$, -OH, -OCH$_3$, -CH=CH$_2$, -SH, -PH$_2$, -F, -Cl and -CHO) were chosen as model compounds, as they potentially exhibit higher $k'_A$ values compared to primary and secondary ones. By calculating the EF for $k'_A$ values of C(CH$_3$)$_2$(CH$_2$($X$))OOH relative to the reference compound C(CH$_3$)$_3$OOH, we found that all nine substituents reduce the $k'_A$ (Fig. 3e). Thus, we conclude that substitutes at C$_\beta$ site could hinder the acid-catalyzed hydrolysis. The unfeasible acid-catalyzed hydrolysis of them aligns with previous experimental results that non-α-substituted monoterpene-derived organic peroxides exhibit greater persistence in aqueous environments (Zhao et al., 2022).

Based on the calculated log $k_A$ values and the $\sum\sigma^*$ constants (Table S2), QSAR models were developed for α- and β-substituted ROOH model compounds. As shown in Fig. S15, a strong linear correlation was established for α-substituted compounds, including CH$_2$($X$)OOH, CH(CH$_3$)($X$)OOH, C(CH$_3$)$_2$($X$)OOH, and C(CH$_3$)$_2$(O$Y$)OOH, with the equation as log $k_A$ = -11.555 $\sum\sigma^*$ + 19.251 ($n$ = 21, $R^2$ = 0.9083) after excluding compounds with unknown $\sum\sigma^*$ and those with substituents -CH=CH$_2$ and -CHO. The poor performance in describing the effects of substituents -CH=CH$_2$ and -CHO is likely because $\sum\sigma^*$ primarily characterizes the inductive effects of substituents (Perrin et al., 1981), while its description of conjugation effects is inadequate. A relatively weaker fit to the QSAR plot (log $k_A$ = -3.499 $\sum\sigma^*$ - 6.437 ($n$ = 9, $R^2$ = 0.7446)) was found for compounds including C(CH$_3$)$_3$OOH and β-substituted compounds C(CH$_3$)$_2$(CH$_2$($X$))OOH (Fig. S16). Nonetheless, a negative correlation was found in both QSAR models. This indicates that the weaker the inductive effects of substituents (i.e., the smaller the $\sum\sigma^*$), the higher the $k_A$, which is consistent with the trend explained by the stability order of the formed carbocation intermediates discussed earlier.

### 3.3 Acid-catalyzed hydrolysis of atmospheric ROOHs

To evaluate the applicability of the structure-activity relationship derived from model compounds to more complicated atmospheric ROOHs, we investigated the acid-catalyzed hydrolysis of eight ROOHs derived from isoprene (Wennberg et al., 2018), α-pinene (Zhang et al., 2017; Claflin et al., 2018), trimethylamine (Kjaergaard et al., 2023; Ma et al., 2021), and dimethyl sulphide (Berndt et al., 2019) using the screened $H^+(H_2O)_2$ model via the DFT calculations (Figs. S17-S24). These ROOHs feature distinct substituents at the $C_\alpha$ site, including $-N(CH_3)_2$, $-N(CH_3)(CHO)$, $-OH$, $-CH=CH_2$, $-SCH_3$, $-SCHO$, and

$-CHO$, and unsubstituted. As shown in Fig. 5, the order of $k'_A$ values for these ROOHs at the same pH (3.8 or 0.9) according to their substituents is as follows: $-N(CH_3)_2 > -OH > -N(CH_3)(CHO) >$ unsubstituted $> -CH=CH_2 > -SCH_3 > -CHO > -SCHO$, similar to the trends observed in model compounds in three aspects. First, a $-N(CH_3)_2$ or $-OH$ substituent at the $C_\alpha$ site of trimethylamine-$OOH_{(1)}$ or α-pinene-$OOH_{(1)}$ results in their high $k'_A$ values, with $\tau_{1/e}$ values less than 2 d (pH 3.8) and 3.6 min (pH 0.9), respectively. Second, the introduction of a $-CHO$ group to the N atom in the $-N(CH_3)(CHO)$ substituent of

trimethylamine-$OOH_{(2)}$ leads to a remarkable reduction in $k'_A$ values, with $\tau_{1/e}$ values exceeding 45 d, even at pH 0.9 (Table S4). The possible reaction of trimethylamine-$OOH_{(2)}$ initiated by the hydrolysis of $-N(CH_3)(CHO)$ group is not considered in this work (Zhang et al., 2015). Moreover, the attachment of a $-CHO$ group to the S atom in the $-SCHO$ substituent leads the corresponding dimethyl sulfide-$OOH_{(2)}$ to exhibit the lowest $k'_A$ values, even lower than that of α-pinene-$OOH_{(3)}$ featuring a $-CHO$ substituent. Finally, the $-CH=CH_2$ and $-SCH_3$ substituents contribute to the relatively low $k'_A$ values of isoprene-OOH

and dimethyl sulfide-OOH, respectively. Although the existence of enhancing substituents, their $k'_A$ values is lower than the unsubstituted α-pinene-$OOH_{(2)}$, which is attributed to the reducing effect of an additional $-OH$ substituent at the $C_\beta$ site for isoprene-OOH, and the primary $C_\alpha$ of sulfide-OOH compared to tertiary $C_\alpha$ of α-pinene-$OOH_{(2)}$. The reproduction of the substituents effect trend on $k'_A$ values in atmospheric ROOHs demonstrates that the structure-activity relationship derived from model ROOHs can be effectively extended to predict the $k_A$ values of structurally diverse ROOHs.

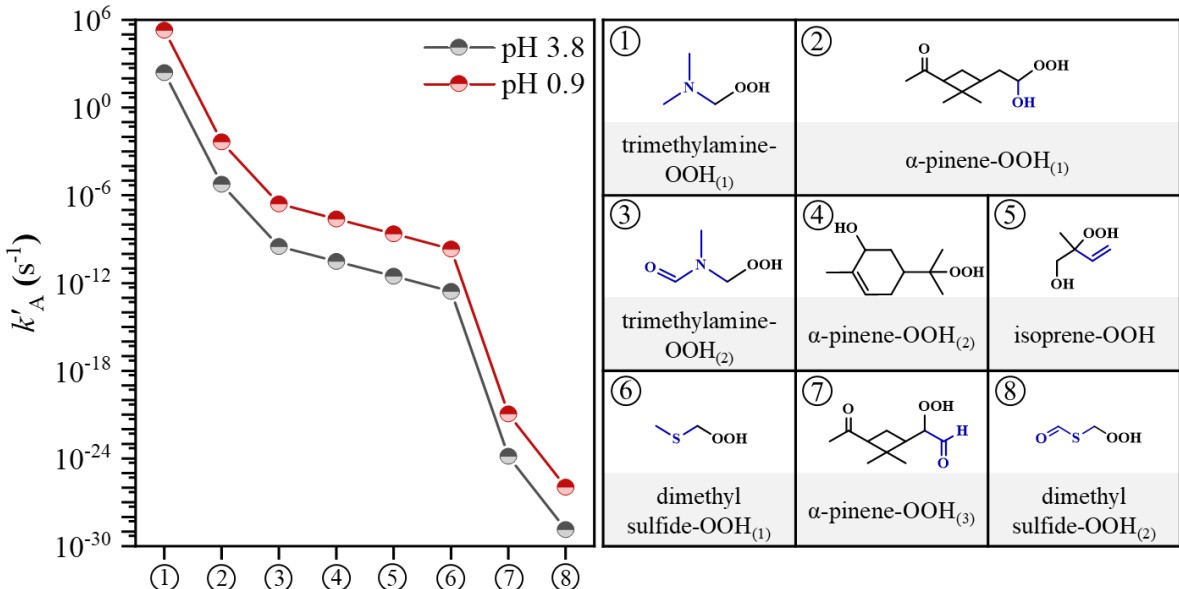

**Figure 5. Calculated pseudo-first-order rate constants ($k'_A$) of eight atmospheric ROOHs under two selected pH values. pH 3.8 and pH 0.9 represent typical aerosol conditions in inland China and the Southeastern United States, respectively.**

In the aerosol aqueous phase, anions such as nitrate ($NO_3^-$) and sulfate ions ($SO_4^{2-}$) are important components in addition to $H_2O$. These anions can either directly nucleophilically attack the $C_\alpha$ of ROOHs or react with the formed carbocation intermediates to produce organic nitrates and organic sulfates (Figs. S3-S6 and S17-S24). Understanding the competition between acid-catalyzed hydrolysis and esterification reactions of atmospheric ROOHs is crucial for elucidating their transformation products in ambient aerosols. Herein, we use $C_{13}$ α-AH, $C_{12}$ α-AH$_{(1)}$, $C_{12}$ α-AH$_{(2)}$, and $C_{10}$ α-HH as examples to elucidate the competition between their acid-catalyzed hydrolysis and esterification reactions. As shown in Figs. S3-S6, reactions of their derived carbocations with $H_2O$ and $NO_3^-$ are endothermic, while those with $SO_4^{2-}$ are barrierless and exothermic. Although the formation of hemiacetals/geminal diols through reactions with $H_2O$ is thermodynamically unfavorable, these species can subsequently be transformed into lactols (Qiu et al., 2020b; Enami, 2022; Hu et al., 2022). Therefore, competitive reactions with $H_2O$ and $SO_4^{2-}$ must be considered. If the concentrations of $H_2O$ and $SO_4^{2-}$ are equal, carbocations preferentially react with $SO_4^{2-}$ at near diffusion-controlled rate constants (~ $10^{10}$ L mol$^{-1}$ s$^{-1}$) to form organic sulfates. In fact, the distribution of products, including hemiacetals/geminal diols, transformed lactols, and organic sulfates in aerosols, is determined by the second-order rate constants for carbocation reactions with $H_2O$ and the variable concentrations of $H_2O$ and $SO_4^{2-}$. The formation mechanisms of organic sulfates have been a hot research topic in atmospheric chemistry. To date, a variety of formation mechanisms have been proposed; among them, the acid-catalyzed ring-opening of epoxides has been widely adopted to explain organic sulfates formation in acidic sulfate aerosol (Surratt et al., 2010; Lin et al., 2012; Cooke et al., 2024). Nevertheless, existing mechanisms still cannot fully explain the observed ambient abundance of organic sulfates. Here we present a plausible carbocation-mediated organic sulfates formation mechanism that occurs during the aqueous-phase transformation of ROOHs in the presence of $H^+$ and $SO_4^{2-}$. However, our structure-activity relationship analysis indicates that,

among our studied ROOHs, only those with $C_\alpha$ substituents such as -OH, -OCH$_3$, -NH$_2$, and -N(CH$_3$)$_2$, can effectively generate the corresponding carbocation intermediates on the lifetime scale of aerosols under the considered pH conditions (pH 3.8 and 0.9). In the atmosphere, such ROOHs primarily derive from the reactions of water, alcohols, or ammonia with CIs produced via ozonolysis of unsaturated hydrocarbons or from ·OH oxidation of tertiary amines (Wang et al., 2023; Li et al., 2024; Kjaergaard et al., 2023). Therefore, we speculate that the proposed carbocation-mediated mechanism would be helpful to explain organic sulfates formation in SOA derived from the aforementioned reactions, whereas its applicability to other sources such as ·OH/NO$_3$· oxidation of volatile organic compounds (Surratt et al., 2008) could be limited unless the precursor compounds contain activating substituents that support the formation of the α-substituted ROOHs mentioned above.

## 4 Conclusions

This study demonstrates that combining the protonated water cluster H$^+$(H$_2$O)$_2$ model with the M06-2X/6-311++G(3df,2pd)//M06-2X/6-31+G(d,p) method can accurately predict the acid-catalyzed hydrolysis kinetics of ROOHs. This approach provides a reliable framework for predicting the reaction kinetics of other atmospheric ROOHs. Furthermore, the study identifies new functional groups, including -NH$_2$, -N(CH$_3$)$_2$, -OH, -OCH$_3$, -CH=CH$_2$, -SH, and -PH$_2$, substituted at the $C_\alpha$ site of the -OOH group, which can enhance the acid-catalyzed hydrolysis kinetics of ROOHs. Notably, the newly identified -NH$_2$ and -N(CH$_3$)$_2$ substituents exhibit a greater enhancing effect than the well-documented -OH and -OCH$_3$ substituents. Contrary to the assumptions based on chemical intuition, it is clarified that not all nitrogen- or oxygen-containing substituents at the $C_\alpha$ site of the -OOH group can equally enhance acid-catalyzed hydrolysis of ROOHs, and the effects depend on the substituents attached to the O or N atoms. Importantly, the structure-activity relationship for commonly encountered substituents was elucidated for the acid-catalyzed hydrolysis kinetics of ROOHs, providing guidelines for qualitatively assessing the feasibility of acid-catalyzed hydrolysis of atmospheric ROOHs in aerosol water. The predicted kinetic data can be incorporated into 3-D chemical models to better simulate the atmospheric aqueous-phase chemistry of ROOHs. Meanwhile, we also emphasize that future work focusing on the standards synthesis of structurally diverse ROOHs would be very helpful for the laboratory validation of our findings.

This study also reveals that carbocation intermediates are formed during the acid-catalyzed hydrolysis of some ROOHs. The formed carbocation could further react with anions and organics, which are abundant in ambient aerosols besides water. Our preliminary studies indicate that the reaction of the formed carbocation with abundant SO$_4^{2-}$ is barrierless and exothermic, leading to the formation of less-volatile organic sulfates. Meanwhile, these carbocations could also mediate the formation of oligomers and N-heterocycles according to the aqueous-phase carbocations chemistry of the α-dicarbonyls (Ji et al., 2020; Shi et al., 2024; Zhang et al., 2022). These findings emphasize the importance of investigating the impact of acid-catalyzed transformations of ROOHs on aerosol composition and properties. Additionally, the aqueous-phase transformations of atmospheric ROOHs with multiple hydrolysable functional groups should be further investigated.

**Data availability.** All data were available in the main text or supplementary materials. The other relevant data are available upon request from the corresponding authors.

**Author contributions.** HX and RY contributed to conceiving the idea, editing, and revision. QZ contributed to conceiving the idea, performing DFT calculations, analyzing results and interpreting data, and writing the original draft. FM, HZ, QX, XW, and JC contributed to editing and revision. All authors read and approved the final manuscript.

**Competing interest.** The authors declare no conflict of interest.

**Financial support.** This work was supported by the National Natural Science Foundation of China (22176022, 22406017, 22206020, 22236004).

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
