# Peer review of "Acid-catalyzed hydrolysis kinetics of organic hydroperoxides: Computational strategy and structure-activity relationship"

_EGUsphere, 2025_

## Author Comment (AC1)

**Responses to Reviewers' Comments on Manuscript egusphere-2025-1662**

(Acid-catalyzed hydrolysis kinetics of organic hydroperoxides: Computational strategy and structure-activity relationship)

*Reviewer 1*

==Reviewer:== **General comment**

*The authors present a computational study investigating the acid-catalyzed hydrolysis rate constants of various hydroperoxides. In a first step, a proton model is screened and tested against experimentally derived hydrolysis rate constants and in a second step, the hydrolysis of a variety of hydroperoxides are investigated and discussed for typical atmospheric conditions. The study aims to overcome limitations of limited availability of authentic standards by using computational methods, which delivers an important contribution to the understanding of the fate of hydroperoxides in the atmosphere. As the reaction also leads to hydrogen peroxide formation, the findings of this study have large implications for the oxidant budget in the atmosphere. The study is well written with a clear flow and logic. I have, however, some concerns about the wide application of a system that was tested for quite narrow conditions. I recommend publication once a few issues have been addressed.*

==Response:== Thanks for the comments. We have revised the manuscript to enhance the quality.

==Reviewer:== **Special Suggestions and Comments**

==Reviewer:== 1) *The authors test their proton model on four compounds that are structurally very similar and all derived from the aqueous-phase ozonolysis of α-pinene with different alcohols as reaction partners of the corresponding Criegee-intermediates. The compounds tested however, span a much wider variety including functional groups such as -NH₂, -PH₂, -SH and -CH=CH₂ and compounds from much different precursors, such as isoprene and DMS. This might lead to significant uncertainties that should be discussed in more detail. Furthermore, hydroperoxides formed in the gas phase might be structurally quite different, as isomerization*

*reactions are expected to be more pronounced.*

**Response:** Thanks for the comment. We acknowledge that the four selected ROOHs ($C_{13}$ α-AH, $C_{12}$ α-AH$_{(1)}$, $C_{12}$ α-AH$_{(2)}$, and $C_{10}$ α-HH) are structurally similar. This was primarily due to the current lack of experimental data for ROOHs containing substituents such as -NH$_2$, -PH$_2$, -SH, and -CH=CH$_2$. Nonetheless, our results show that nearly all studied ROOHs share identical reaction sites (i.e., the -OOH group and its adjacent $C_\alpha$ atom) and follow analogous reaction pathways to $C_{13}$ α-AH, $C_{12}$ α-AH$_{(1)}$, $C_{12}$ α-AH$_{(2)}$, and $C_{10}$ α-HH. This mechanistic consistency suggests that, despite differences in their carbon skeletons and substituents, including those of gas-phase formed ROOHs, the key protonation and subsequent reactions remain localized at the $C_\alpha$-OOH site, which is a common structural feature among ROOHs. Thus, we believe the screened protonated water cluster model can be reasonably extended to investigate the acid-catalyzed hydrolysis of diverse ROOHs with acceptable uncertainty. We have added the corresponding discussion in the revised manuscript (see lines 168 – 174).

**Reviewer:** 2) *More to this point, there are some hydroperoxides also commercially available and synthetic procedures have been published for others. Although I recognize that determining the hydrolysis rate constants for these compounds might not be within the scope of this study, the limitations of this procedure should be discussed.*

**Response:** Thanks for the comment. Although some hydroperoxides are commercially available or synthetically accessible, standard compounds remain limited, especially for ROOHs with heteroatom substituents (e.g., -NH$_2$, -PH$_2$, -SH). This limits experimental validation of our computational results. Nevertheless, we emphasize the importance of the synthesis and characterization of a broader range of ROOHs to support future validation. Please see lines 329 - 331 in the revised manuscript.

**Reviewer:** 3) *The authors apply the model to atmospheric ROOHs described in the literature. The cited study corresponding to DMS oxidation shows in fact, that instead of CH₃SCH₂OOH discussed in this study, a pronounced isomerization step mainly leads to the formation of*

*CHOSCH$_2$OOH, again a more functionalized compound. I suggest including this compound in the list of tested compounds.*

**Response:** We appreciate the reviewer's helpful suggestion. CHOSCH$_2$OOH has now been included in our analysis and discussed in the revised manuscript. Please see Fig. 5, section 3.3 in the revised manuscript, and Fig. S24 in the SI.

**Reviewer:** 4) *In Line 193 ff., the authors discuss the effect of functional groups in their findings and trace it back to a stabilization of the intermediate step. Although I can support the analysis, I want to point out, that the reaction pathway and the nature of the intermediate step is determined by the method the authors applied. This trend is caused by the input parameters for this analysis, which I think should be reflected by the discussion.*

**Response:** Thanks for the comment. To assess the robustness of our results, we evaluated the influence of different computational methods on the reaction free-energy barriers using C(CH$_3$)$_2$($X$)OOH ($X$ = NH$_2$, OH, OCH$_3$, CH=CH$_2$, SH, and PH$_2$) as representative compounds. As illustrated in Fig. R1, although the absolute values of the free-energy barriers and reaction free energies vary between the ωB97X-D/6-311++G(3df,2pd)//ωB97X-D/6-31+G(d,p) method and the M06-2X/6-311++G(3df,2pd)//M06-2X/6-31+G(d,p) method, the overall trend across ROOHs with different substituents remains consistent. This confirms that our conclusions are not strongly dependent on the specific method applied. The corresponding discussion has been added in the revised manuscript, please see lines 226-227 in the revised manuscript, and Fig. S13 in the SI.

[Figure]

**Figure R1.** Calculated free-energy surfaces for carbocation formation during the acid-catalyzed hydrolysis of $C(CH_3)_2(X)OOH$, where $X$ = $NH_2$, OH, $OCH_3$, $CH=CH_2$, SH, or $PH_2$. Free energies are computed at (a) M06-2X/6-311++G(3df,2pd)//M06-2X/6-31+G(d,p) and (b) ωB97X-D/6-311++G(3df,2pd)//ωB97X-D/6-31+G(d,p) levels with the SMD solvation model. R, TS, and IM denote the reactant, transition state, and intermediate, respectively.

**Reviewer:** 5) *Similarly, in lines 254, the authors claim, that the results prove that the SAR can be applied to atmospheric ROOHs, but as far as I understand there is no proof that the hydrolysis constants derived are valid for atmospheric conditions. I suggest to rephrase that section to better reflect that point.*

**Response:** Thanks for the comment. We fully agree with the reviewer and have revised the relevant description "…can be effectively extended to atmospheric ROOHs" to "… can be effectively extended to predict the $k_A$ values of structurally diverse ROOHs" accordingly. Please

see line 299 in the revised manuscript.

**Reviewer:** 6) *In the conclusion, the authors introduce the reaction with sulfate of the intermediate carbocation that was previously not mentioned. This might have large implications in the atmosphere and I would suggest including this reaction pathway as well as potentially the reaction with nitrate in the discussion section.*

**Response:** We appreciate the reviewer's insightful suggestion. In response, we have expanded the discussion to include the potential reactions of the carbocation intermediate with $H_2O$, $NO_3^-$, and $SO_4^{2-}$. Please see lines 303 - 316 in the revised manuscript.

---

## Author Comment (AC2)

**Responses to Reviewers' Comments on Manuscript egusphere-2025-1662**

(Acid-catalyzed hydrolysis kinetics of organic hydroperoxides: Computational strategy and structure-activity relationship)

*Reviewer 2*

**Reviewer:** **General comment**

*Organic hydroperoxides are formed during the photochemical oxidation of organic compounds and represent a non-negligible fraction of the atmospheric aerosol mass. This study introduces a computational strategy to evaluate the hydrolysis rate constants of organic hydroperoxides and uses this strategy to explore the dependence of hydrolysis rate on molecular structure of hydroperoxides and also on medium acidity. The study is well executed, the results are mostly sound, and the manuscript is well written. I believe that it can be published subject to several changes as described below.*

**Response:** Thanks for your insightful comments. We have made point-by-point responses in the following section.

**Reviewer:** **Special Suggestions and Comments**

**Reviewer:** 1). *The introduction section must be expanded a bit to make it easier to understand the chemistry of the processes under consideration. The two possible hydrolysis mechanisms, concerted single-step and two-step depicted in Figure 2b, must be introduced in the introduction section. Adding a variant of Figure S7 to the introduction section would also be beneficial. The meaning of the proton model needs to be defined on the first occurrence. Is it "proton model" or "proton donor model"?*

**Response:** Thanks for the comment. We have expanded the introduction to include a clearer explanation of the acid-catalyzed hydrolysis mechanisms. Please see lines 49 - 53 in the revised manuscript. Regarding the term used, based on literature (Agmon et al., 2016; Schran et al., 2020; Fournier et al., 2015), "protonated water clusters, $H^+(H_2O)_n$" is consistently used to

describe molecular clusters formed between $H^+$ and water molecules via hydrogen bonding. Thus, we have replaced "proton model" with "protonated water cluster model" or "$H^+(H_2O)_n$ model" throughout the manuscript, adding the definition at its first occurrence. Please see lines 14 - 16, 66 - 71, 77, 101 - 109, 128, 136, 138, 318 in the revised manuscript.

**Reviewer:** 2). *A highly non-monotonic dependence of the hydrolysis rate constant on the size of the proton model (the number of water molecules clustered around hydronium ion) is observed. One and three water molecules produce a nearly similar effect while with two water molecules the rate is significantly slower. This effect needs to be discussed, and its origin must be established.*

**Response:** We thank the reviewer for pointing out this issue. We have double-checked the non-monotonic dependence of the hydrolysis rate constants on the size of the protonated water cluster models. We found that in the original version of the manuscript, the ROOH molecule first replaced one $H_2O$ molecule in the protonated water cluster model $H^+(H_2O)_4$ (i.e., $H_3O^+(H_2O)_3$), differing from the direct protonation of ROOH by the other three protonated water cluster models $H^+(H_2O)_{1-3}$ (i.e., $H_3O^+(H_2O)_{0-2}$) during the reaction. This discrepancy could be the reason for the non-monotonic dependence of the hydrolysis rate constants. After careful consideration, we recognized that we should not break the protonated water cluster model. Therefore, we recalculated the rate constants for the reaction involving direct protonation of ROOH by $H^+(H_2O)_4$ and found that the updated hydrolysis rate constants decrease monotonically with increasing cluster size ($n$ = 1 to 4) for the reactions of ROOHs with $H^+(H_2O)_n$. We have revised the corresponding part in the manuscript accordingly. Please see line 141, Fig. 1 in the revised manuscript, and Figs. S3-S6 in the SI.

**Reviewer:** 3). *The largest enhancement in the hydrolysis rate is reported for the nitrogen-containing hydroperoxides, and it is assumed that the amino groups remain unprotonated. This is grossly incorrect, as in the pH range considered in this study the fraction of the free, unprotonated form of -NH₂ and -N(CH₃)₂ will be exceedingly small. To calculate the correct*

*lifetime of these N-containing hydroperoxides, the fraction of the unprotonated form must be evaluated based on pK_b. The latter can be estimated based on the data available for similar compounds in the literature or calculated explicitly from the Gibbs free energy of the amino group protonation evaluated by DFT.*

**Response:** We appreciate the comment and agree that $-NH_2$ and $-N(CH_3)_2$ groups are predominantly protonated (i.e., $-NH_3^+$ and $-NH(CH_3)_2^+$) under the pH conditions considered in this study. However, the rapid reaction of ROOHs with the unprotonated $-NH_2$ and $-N(CH_3)_2$ species may shift the equilibrium $-NH_3^+$ $(-NH(CH_3)_2^+)$ + $H_2O$ $\leftrightarrow$ $-NH_2$ $(-N(CH_3)_2)$ + $H_3O^+$ toward the unprotonated form. Recent studies (Ji et al., 2020; Shi et al., 2024) have reported fast reactions between small α-dicarbonyls and amines or ammonia in acidic aqueous solutions, with corresponding products detected experimentally. This indicates the ROOHs with unprotonated $-NH_2$ and $-N(CH_3)_2$ can still participate in the reaction. In addition, Enami et al. (2010) found that trimethylamine remains largely unprotonated at the air-water interface even at pH 4.0. These findings suggest that both aqueous-phase and interfacial transformations of α-$NH_2/N(CH_3)_2$-substituted ROOHs in unprotonated form are plausible, highlighting their significance in atmospheric aqueous-phase chemistry. We have incorporated this discussion in the revised manuscript. Please see lines 209 - 216 in the revised manuscript.

**Reviewer:** 4). A free carbocation is shown in Figure S7. Is using an implicit solvent model sufficient to stabilize this carbocation? How much would the reaction energetics change if this carbocation is stabilized explicitly, e.g., by hydration?

**Response:** Thanks for the comment. To evaluate hydration effects on the reaction energetics, we conducted additional calculations using $C(CH_3)_2(OCH_3)OOH$ as a model compound. We identified the most stable monohydrated configurations for both the reactant (R, $C(CH_3)_2(OCH_3)OOH$) and its corresponding carbocation intermediate (IM), and compared the reaction energetics with those obtained using only the implicit solvation model. As shown in Fig. R1, adding one explicit $H_2O$ molecule to R and IM increases the reaction energetics by only 1.46 or 1.74 kcal mol$^{-1}$, respectively, indicating that hydration has a minimal effect on the

overall reaction energetics. This discussion has been added to the revised manuscript. Please see lines 227 - 229 in the revised manuscript and Fig. S14 in the SI.

[Figure]

**Figure R1. Changes in reaction energetics for carbocation intermediate formation during acid-catalyzed hydrolysis of C(CH₃)₂(OCH₃)OOH under (a) implicit solvation only vs. (b) explicit monohydration. Free energies (*G*) computed at SMD/M06-2X/6-311++G(3df,2pd)//M06-2X/6-31+G(d,p) level. R and IM denote the reactant and carbocation, while R ·· H₂O and IM ·· H₂O are their monohydrated forms.**

**Reviewer:** 5). *Having read the title, I assumed that the paper will eventually present some kinds of quantitative structure-reactivity relationship. It did not and it is a pity, as a pretty large dataset has been produced. Is it possible to relate the rate constant with some parameters of the substituents, e.g., similar as in the Hammett equation? This would be very beneficial for the modeling studies.*

**Response:** Thanks for the suggestion. We agree a quantitative structure-reactivity relationship would be highly valuable. Similar to Hammett constants the reviewer mentioned, Taft ($\sigma^*$) constants are widely used to quantify substituent effects in aliphatic systems (Perrin et al., 1981). We have now established correlation equations linking $\sigma^*$ to the logarithms of the acid-catalyzed hydrolysis second-order rate constants (log $k_A$) for the selected ROOH model compounds. For more detailed information, please see lines 118 - 124, 268 - 278 in the revised manuscript and Figs. S15-S16 in the SI.

**Reviewer:** 6). L35: more hydrophilic HYDROperoxide groups

**Response:** Thanks for the comment. We have revised it. Please see line 38.

**Reviewer:** 7). L45: remove "extremely"

**Response:** Thanks for the comment. We have revised it. Please see line 47.

**References**

Agmon, N., Bakker, H. J., Campen, R. K., Henchman, R. H., Pohl, P., Roke, S., Thämer, M., and Hassanali, A.: Protons and hydroxide ions in aqueous systems, Chem. Rev., 116, 7642-7672, 10.1021/acs.chemrev.5b00736, 2016.

Enami, S., Hoffmann, M. R., and Colussi, A. J.: Proton availability at the air/water interface, J. Phys. Chem. Lett., 1, 1599-1604, 10.1021/jz100322w, 2010.

Fournier, J. A., Wolke, C. T., Johnson, M. A., Odbadrakh, T. T., Jordan, K. D., Kathmann, S. M., and Xantheas, S. S.: Snapshots of proton accommodation at a microscopic water surface: Understanding the vibrational spectral signatures of the charge defect in cryogenically cooled $H^+(H_2O)_{n=2–28}$ clusters, J. Phys. Chem. A, 119, 9425-9440, 10.1021/acs.jpca.5b04355, 2015.

Ji, Y., Shi, Q., Li, Y., An, T., Zheng, J., Peng, J., Gao, Y., Chen, J., Li, G., Wang, Y., Zhang, F., Zhang, A. L., Zhao, J., Molina, M. J., and Zhang, R.: Carbenium ion-mediated oligomerization of methylglyoxal for secondary organic aerosol formation, Proc. Natl. Acad. Sci. U. S. A., 117, 13294-13299, 10.1073/pnas.1912235117, 2020.

Perrin, D. D., Dempsey, B., and Serjeant, E. P.: $pK_a$ prediction for organic acids and bases, Champman and Hall: New York, 1981.

Schran, C., Behler, J., and Marx, D.: Automated fitting of neural network potentials at coupled cluster accuracy: Protonated water clusters as testing ground, J. Chem. Theory Comput., 16, 88-99, 10.1021/acs.jctc.9b00805, 2020.

Shi, Q., Gao, L., Li, W., Wang, J., Shi, Z., Li, Y., Chen, J., Ji, Y., and An, T.: Oligomerization mechanism of methylglyoxal regulated by the methyl groups in reduced nitrogen species: Implications for brown carbon formation, Environ. Sci. Technol., 58, 1563-1576, 10.1021/acs.est.3c05983, 2024.

---

## Author Comment (AC3)

**Responses to Reviewers' Comments on Manuscript egusphere-2025-1662**

(Acid-catalyzed hydrolysis kinetics of organic hydroperoxides: Computational strategy and

structure-activity relationship)

*Reviewer 3*

Reviewer: **General comment**

*The manuscript "Acid-catalyzed hydrolysis kinetics of organic hydroperoxides: Computational strategy and structure-activity relationship" presents an updated DFT-calculated proton model probing the structure activity relationships of organic peroxides, varied functional groups and acid-catalyed hydrolysis. The manuscript incorporates empirical data for four initial compounds and once the model fits with observed values, expands the model to include several functional groups. The manuscript is clearly written and describes the work. However, there are several locations where clarification or additional information is needed before the manuscript is ready for publication. I recommend for publication after addressing the following questions and corrections:*

Response: Thanks for the comment. We have carefully revised the manuscript to improve its overall quality.

Reviewer: **Special Suggestions and Comments**

Reviewer: 1). *The abstract (line 13) uses of the word solved and this is a broad assertion, since the model is built using a sparse set of empirically derived data points. Are you asserting that this method replaces the need for all authentic ROOH standards and can be used in lieu of authentic ROOH standards? If not, I recommend changing this word. Additionally, (line 17) define $C_\alpha$ prior to first use and clarify if the 52 ROOHs include the four ROOH stated in the previous sentence because it is unclear where the total 52 compounds originate.*

Response: Thanks for the suggestion. We have revised the manuscript accordingly: (1) The phrase "solved this issue" has been replaced with "addressed this limitation"; (2) $C_\alpha$ is clearly

defined at its first mention as the carbon atom directly bonded to the -OOH group; (3) We have clarified the composition of the selected ROOHs. Following suggestions from Reviewer 1, we added one DMS-derived ROOH, updating the total number from 52 to 53. The final dataset includes 45 model compounds and 8 atmospherically relevant species, excluding 4 ROOHs used for protonated cluster model selection. Please see lines 13, 17 - 19 in the revised manuscript.

**Reviewer:** 2). *The introduction needs to be expanded to further detail the work this manuscript is building on. Specifics include: (line 31) clarify what is meant by lack of kinetic data in this sentence, there are many types of kinetics data beyond acid-catalyzed hydrolysis. Similar to the abstract, (line 68) please clarify if 45 ROOH model compounds were used or if 52 compounds were used as is stated in the abstract and (line 68) define $C_\alpha$ and $C_\beta$ prior to first use. The methods section is well written and straight forward, however the empirical values are not referenced explicitly, (line 100-101) please clarify if $C_{13}$ α-AH, $C_{12}$ α-AH$_{(1)}$, $C_{12}$ α-AH$_{(2)}$, and $C_{10}$ α-HH were chosen based on a specific reference, i.e., Hu 2022. Figure 1 in the results and discussion section should be altered to clearly designate the empirically derived values. Currently, the experimental values are currently orange and difficult to see on the graph. Please consider changing to a different contrasting color, such as black, so the empirical data will be distinct from the model data. Section 3.3 Acid-catalyzed hydrolysis of atmospheric ROOHs needs to be expanded. Specifically, (lines 239-241) please clarify if these seven compounds have experimental kinetics data and if they were used in the model.*

**Response:** Thanks for the comment. We have revised the manuscript accordingly: (1) In the introduction, we clarify that the "lack of kinetic data" refers specifically to aqueous-phase processes, including acid-catalyzed hydrolysis, esterification, oxidation, photosensitization, and oligomerization. Please see lines 33 - 34; (2) We specified the ROOH dataset composition (45 model compounds and 8 atmospheric species) and defined $C_\alpha$ (the carbon atom directly bonded to the -OOH group) and $C_\beta$ (the carbon attached to $C_\alpha$) at their first mention. Please see lines 72, 74 - 76; (3) We added the reference to support the selection of $C_{13}$ α-AH, $C_{12}$ α-AH$_{(1)}$,

$C_{12}$ α-AH$_{(2)}$, and $C_{10}$ α-HH in the methods. Please see line 108; (4) Fig. 1 was revised to improve clarity by changing the color of experimental data markers to dark grey. Please see Fig. 1; (5) In Section 3.3, we clarified that the hydrolysis rate constants for the eight atmospheric ROOHs were predicted using the screened $H^+(H_2O)_2$ model via the density functional theory calculations, and to the best of our knowledge, their experimental hydrolysis rates remain unavailable. Please see line 283 in the revised manuscript.

==Reviewer:== 3). *The supplemental information needs to be more thoroughly explained. Figure S1 needs to have the chemical formulas and molecular weights for the molecules listed (a-d) beneath the molecules. For figures S3-S6, please clarify how the products were determined, i.e., the reactions including $H_2O$, $NO_3^-$, $(SO_4)^{2-}$. Those products appear to be uniformly formed regardless of case. In addition, the SI must include a reference for the four compounds with empirical data need to be referenced here.*

==Response:== Thanks for the comment. We have added the chemical formulas and molecular weights for the molecules in Fig. S1. For Figs. S3-S6, the reaction product of carbocation intermediates with $H_2O$, $NO_3^-$, and $(SO_4)^{2-}$ were assessed based on their reaction thermodynamics and kinetics information. A more detailed discussion has been added in the revised manuscript. Additionally, the relevant references have also been cited in the SI. Please see lines 303 - 316 in the revised manuscript, and Fig. S1, references in the SI.

---

## Author Comment (AC4)

**Responses to Reviewers' Comments on Manuscript egusphere-2025-1662**

(Acid-catalyzed hydrolysis kinetics of organic hydroperoxides: Computational strategy and structure-activity relationship)

*Reviewer 4*

Reviewer: **General comment**

*This study focuses on improving our representation of acid-catalyzed hydrolysis of ROOHs. The authors appear to have a strong understanding of the computational models employed in this study and do a good job of explaining the chemical reasoning behind the modeled behavior. Additionally, the figures are clear, helpful, and well-made. I recommend this paper for publication after very minor changes.*

Response: We sincerely appreciate your positive comment and have further refined the manuscript based on your valuable suggestions.

Reviewer: **Special Suggestions and Comments**

Reviewer: 1) *In Section 2.2, it would be helpful to have more background on why those 4 compounds were chosen. In the introduction you describe the importance of alpha-HHs and alpha-AHs, and it would be good in Section 2.2 to include a brief description and/or references to explain why these specific 4 were chosen.*

Response: Thanks for the comment. The four ROOHs were chosen due to the availability of experimental kinetic data under different pH conditions. These compounds also represent typical structures of α-AHs and α-HHs highlighted in the introduction. We have added these explanations and supporting references in Section 2.2. Please see lines 107 - 108 in the revised manuscript.

Reviewer: 2) *Consider expanding the discussion of future work in the Conclusions section. Do you think that more modelling studies, laboratory validation studies, or both would be helpful*

*to expand on and utilize this work?*

Response: We appreciate the reviewer's insightful comment and fully agree that both further modelling and laboratory validation studies are necessary to expand our findings. Accordingly, we have developed quantitative structure-activity relationship models linking Taft ($\sigma^*$) constants to the logarithms of the acid-catalyzed hydrolysis second-order rate constants (log $k_A$). In the revised manuscript, we emphasize the importance of laboratory validation in the Conclusions section. Please see lines 118 - 124, 268 - 278, 329 - 331 in the revised manuscript, and Figs. S15 and S16 in the SI.

Reviewer: 3) *There are some minor grammatical errors throughout the paper. I recommend having a native English speaker review the paper.*

Response: Thanks for the suggestion. The revised manuscript has been professionally reviewed by a native English speaker to address grammatical issues throughout.

Reviewer: 4) *Specify the pH that was used to calculate the enhancement factors in Figure 3 b-f. Or are these enhancement factors constant across pH?*

Response: Thanks for the comment. The enhancement factors remain constant across different pH values because the predicted acid-catalyzed hydrolysis rate constants of ROOHs show a linear dependence on the concentration of protonated water clusters, and thus on pH. We have added the corresponding description in the revised manuscript (see lines 179 - 180).

---

## Author Response (AR2)

Re: Manuscript egusphere-2025-1662

Dear Prof. Jason Surratt,

Thanks for handling our manuscript and giving insightful comments. All your comments have been addressed in the revised manuscript. We hope that the following responses are satisfying and that the paper can be accepted for publication.

Thanks for your consideration!

Sincerely yours,

Hong-Bin Xie, Rujing Yin

School of Environmental Science and Technology,

Dalian University of Technology,

Linggong Road 2, Dalian 116024,

China

(Acid-catalyzed hydrolysis kinetics of organic hydroperoxides: Computational strategy and structure-activity relationship)

**Editor:** Suggestions and Comments

**Editor:** 1) *Page 13, where you added new text on lines 310-324:*

*You mention the possibility of forming organic nitrates and organic sulfates, but don't cite any prior papers of these being found in aerosols and possibly formed through aqueous-phase reactions. Please include some references to prior work.*

**Response:** Thanks for the comment. We have added the discussion and relevant references citations in the revised manuscript. Please see lines 316 - 329 in pages 13 and 14.

**Editor:** 2) *Related to # 1 above, I think it is important to note the possible formation of organic nitrates and organic sulfates from aqueous-phase chemistry of ROOHs is not a mechanism well represented in current models for these important reservoir species of N and S. Furthermore, is it possible to speculate if any prior reported organic sulfates (especially since they seem more favorable based on your work) in $PM_{2.5}$ analyzed from field studies (e.g., Surratt et al., 2008, J. Phys. Chem. A) could form from this pathway? I understand if you don't want to, but in some of the original work on organic sulfates (such as the one I cite here), it was unclear at that time what the exact sources of the organic sulfate were (despite knowing isoprene and monoterpene oxidation in the presence of acidic sulfate aerosols could yield these). As the authors know, at least for isoprene oxidation, many of these organic sulfates have been shown to form from aqueous-phase chemistry of epoxides in the presence of $H^+$ and $SO_4^{2-}$ (Surratt et al., 2010, PNAS; Lin et al., 2012, ES&T; Cooke et al., 2024, ES&T).*

**Response:** We appreciate the editor's insightful comment regarding the formation of organic sulfates through the aqueous-phase chemistry of organic hydroperoxides (ROOHs). In this study, we present a plausible carbocation-mediated organic sulfates formation mechanism that occurs during the aqueous-phase transformation of ROOHs in the presence of $H^+$ and $SO_4^{2-}$.

However, our structure-activity relationship analysis indicates that, among our studied ROOHs, only those with $C_\alpha$ substituents such as -OH, -OCH$_3$, -NH$_2$, and -N(CH$_3$)$_2$, can effectively generate the corresponding carbocation intermediates on the lifetime scale of aerosols under the considered pH conditions (pH 3.8 and 0.9). In the atmosphere, such ROOHs primarily derive from the reactions of water, alcohols, or ammonia with Criegee intermediates produced via ozonolysis of unsaturated hydrocarbons or from ·OH oxidation of tertiary amines (Wang et al., 2023; Li et al., 2024; Kjaergaard et al., 2023). Therefore, we speculate that the proposed carbocation-mediated mechanism would be helpful to explain organic sulfates formation in SOA derived from the aforementioned reactions, whereas its applicability to other sources such as ·OH/NO$_3$· oxidation of volatile organic compounds (Surratt et al., 2008) could be limited unless the precursor compounds contain activating substituents that support the formation of the α-substituted ROOHs mentioned above. We have added the corresponding discussion in the revised manuscript, please see lines 316 - 329 in pages 13 and 14.

**References**

Kjaergaard, E. R., Moller, K. H., Berndt, T., and Kjaergaard, H. G.: Highly efficient autoxidation of triethylamine, J. Phys. Chem. A, 127, 8623-8632, 10.1021/acs.jpca.3c04341, 2023.

Li, X., Jia, L., Xu, Y., and Pan, Y.: A novel reaction between ammonia and Criegee intermediates can form amines and suppress oligomers from isoprene, Sci. Total Environ., 956, 177389, 10.1016/j.scitotenv.2024.177389, 2024.

Surratt, J. D., Gómez-González, Y., Chan, A. W. H., Vermeylen, R., Shahgholi, M., Kleindienst, T. E., Edney, E. O., Offenberg, J. H., Lewandowski, M., Jaoui, M., Maenhaut, W., Claeys, M., Flagan, R. C., and Seinfeld, J. H.: Organosulfate formation in biogenic secondary organic aerosol, J. Phys. Chem. A, 112, 8345-8378, 10.1021/jp802310p, 2008.

Wang, S., Zhao, Y., Chan, A. W. H., Yao, M., Chen, Z., and Abbatt, J. P. D.: Organic peroxides in aerosol: Key reactive intermediates for multiphase processes in the atmosphere, Chem. Rev., 123, 1635-1679, 10.1021/acs.chemrev.2c00430, 2023.